# Identification and Quantification of a Phytotoxic Metabolite from *Alternaria dauci*

**DOI:** 10.3390/molecules25174003

**Published:** 2020-09-02

**Authors:** Martha Leyte-Lugo, Pascal Richomme, Pascal Poupard, Luis M. Peña-Rodriguez

**Affiliations:** 1Unidad de Biotecnología, Centro de Investigación Científica de Yucatán, 97205 Mérida, Yucatán, Mexico; 2UPRES EA921SONAS, SFR 4207 QUASAV, Université d’Angers, 49045 Angers, France; pascal.richomme@univ-angers.fr; 3UMR 1345 IRHS, SFR 4207 QUASAV, INRAE, Institut Agro, Université d’Angers, 49045 Angers, France; pascal.poupard@univ-angers.fr

**Keywords:** phytotoxins, *Alternaria* leaf blight, *Daucus carota*, *p*-hydroxybenzoic acid, *α*-acetylorcinol

## Abstract

*Alternaria dauci* is the causal agent of *Alternaria* leaf blight (ALB) in carrot (*Daucus carota*) crops around the world. However, to date, *A. dauci* has received limited attention in its production of phytotoxic metabolites. In this investigation, the bioassay-guided isolation of the extract from liquid cultures of *A. dauci* resulted in the isolation of two metabolites identified as *α*-acetylorcinol (**1**) and *p*-hydroxybenzoic acid (**2**), based on their spectroscopic data and results from chemical correlation reactions. Testing of both metabolites in different assays showed an important phytotoxic activity for *p*-hydroxybenzoic acid (**2**) when tested in the leaf-spot assay on parsley (*Petroselinum crispum*), in the leaf infiltration assay on tobacco (*Nicotiana alata*) and marigold (*Tagetes erecta*), and in the immersion assay on parsley and parsnip (*Pastinaca sativa*) leaves. Quantification of the two metabolites in the crude extract of *A. dauci* kept at different times showed that *p*-hydroxybenzoic acid (**2**) is one of the first metabolites to be synthesized by the pathogen, suggesting that this salicylic acid derivative could play an important role in the pathogenicity of the fungus.

## 1. Introduction

The *Alternaria* genus is well-known for its pathogenicity against many economically important crops worldwide [1,2]. It has been reported that this genus produces toxic metabolites, commonly known as phytotoxins, that play a significant role during the infection process [3,4,5,6]. One *Alternaria* species that has received limited attention, particularly in terms of its production of phytotoxic metabolites, is *Alternaria dauci* (Kuhn) Groves & Skolko, recognized as the causal agent of the *Alternaria* Leaf Blight (ALB) disease in cultivated carrots (*Daucus carota*) around the world [7]. The disease appears initially in the leaves, as small greenish-brown lesions of different shapes and sizes; these lesions become larger and turn dark-brown, and are often surrounded by a chlorotic halo [8]. The main effects associated with the disease are the reduction in leaf photosynthetic activity and carbohydrate production, followed by separation of necrotic foliage from the taproot. These effects contribute to a significant reduction in the harvesting efficiency of infected carrots [7].

Previous studies on phytotoxin production by *A. dauci* have reported the identification of some non-host-selective phytotoxins (non-HST′s), including alternariol, alternariol monomethyl ether, and zinniol [5,7,8,9,10,11]. Recently, it was demonstrated that zinniol is not responsible for the phytotoxic effect caused by the organic extract of *A. dauci* [12]; these results, which strongly suggested that the phytotoxic effect of the organic extract of cultures from *A. dauci* was caused by other lipophilic metabolites, were confirmed with the identification of aldaulactone as a new phytotoxin involved in the aggressiveness of *A. dauci* against carrot cells [5]. Given these findings, and as part of our continuing interest in the isolation and identification of phytotoxic metabolites produced by *A. dauci* and their role in the infection process, we wish to report here on the bioassay-guided purification and identification of *α*-acetylorcinol (**1**) and *p*-hydroxybenzoic acid (**2**) as two additional lipophilic phytotoxic metabolites produced by the fungal pathogen.

## 2. Results and Discussion

The bioassay-guided chromatographic purification of the medium polarity (ethyl acetate) fraction from the extract of *A. dauci*, using the leaf-spot assay on parsley leaves as a guide, resulted in the isolation of two phytotoxic metabolites identified as *α*-acetylorcinol (**1**) and *p*-hydroxybenzoic acid (**2**) (Figure 1). The resorcinol derivative **1** showed a single component, with a molecular ion peak at *m*/*z* 166 corresponding to a molecular formula of C_9_H_10_O_3_, when analyzed by gas chromatography–mass spectrometry (GC-MS). Its spectroscopic (^1^H and ^13^C NMR) data coincided with those reported for *α*-acetylorcinol (**1**), the principal metabolite isolated from *Cochliobolus lunata* [13]. Treatment of **1** with iodomethane produced the expected dimethyl ether derivative, which confirmed the presence of two hydroxyl groups in the structure. Alternatively, the IR spectrum of *p*-hydroxybenzoic acid (**2**) showed absorption bands corresponding to hydroxyl (3465 cm^−1^) and carbonyl (1673 cm^−1^) groups, and its GC-MS analysis showed a single component with a molecular ion peak at *m*/*z* 138, corresponding to a molecular formula of C_7_H_6_O_3_. Its spectroscopic data coincided with those reported for **2**, obtained by the oxidative degradation of methylparaben [14,15].

The phytotoxic activity of metabolites **1** and **2** was further evaluated using the leaf infiltration assay in tobacco (*Nicotiana alata*), and marigold (*Tagetes erecta*) leaves (Figure 2) and the leaf immersion assay in parsnip (*Pastinaca sativa*) and parsley (*Petroselinum crispum*) (Figure 3). In the leaf infiltration assay, *p*-hydroxybenzoic acid (**2**) caused a significant phytotoxic effect in both tobacco and marigold leaves at all concentrations, and particularly at 14.5 and 22.0 mM (Figure 2a,b,d,e), while *α*-acetylorcinol (**1**) only caused a minor effect when tested at the highest concentration (22.0 mM) in tobacco leaves (Figure 2c), and proved to be non-phytotoxic when tested on marigold leaves (data not shown). Similarly, *p*-hydroxybenzoic acid (**2**) showed varying levels of phytotoxic activity when tested in the immersion assay using parsnip (necrotic effect at 22.0 mM) and parsley (phytotoxic activity at 14.5 and 22.0 mM) (Figure 3a–c,e,f), while *α*-acetylorcinol (**1**) caused a minimal effect when tested on parsnip (Figure 3d) and none when tested on parsley (Figure 3g).

The benzoic acid derivative *p*-hydroxybenzoic acid (**2**) has been isolated from many natural sources [16,17], including plant species such as oil palm (*Elaeis guineensis*) [18], grapes (*Vitis vinifera*) [19], betel palm (*Areca catechu*) [20], Cuban royal palm (*Roystonea regia*) [21], medlar (*Mespilus germanica*) [22], and black-jack (*Bidens pilosa*) [23]. Furthermore, it is interesting to mention that *p*-hydroxybenzoic acid (**2**) has been reported to accumulate in *Agrobacterium rhizogenes*-induced hairy root cultures of carrots (*D. carota*) [24] and to be produced when carrot cell cultures are exposed to fungal elicitors [25]; this suggests that this benzoic acid derivative might play an important defensive role in carrot cultures, either by itself or as an intermediate in the biosynthetic pathway of other metabolites [25]. However, the fact that *p*-hydroxybenzoic acid (**2**) has been reported as a toxic metabolite at high concentrations [26,27] might explain the phytotoxic activity found in this investigation.

Reports on the production of **2** by fungi are limited, but it has been reportedly obtained from *Alternaria* spp [2], including *A. tagetica* [28]. Additional reports described its production in cultures of the phytopathogens *Epichloë bromicola* (infecting *Elymus tangutorum*) [29] and *Diaporthe gulyae* (isolated from stem cankers of sunflower) [30]. The many biological activities reported for hydroxybenzoic acid (**2**) include phytotoxic activity against *Rumex crispus* [31] and as a growth inhibitor of *Dactylis glomerate* [32]; additionally, it has been reported to have antibacterial [15], antioxidant [33], antifungal [23], antialgal [34], antimutagenic [35], and estrogenic properties [36].

The resorcinol derivative *α*-acetylorcinol (**1**) has also been reported as a secondary metabolite from various *Alternaria* spp [2], including *A. tenuissima* and *A. brassicicola* [37,38], as well as from fungal species including *Cochliobolus lunata* [13], *Stagonospora apocyni* [39], *Verticillium* sp. (an endophyte of *Rehmannia glutinosa*) [40], and *Cladosporium perangustm* FS62 [41]. The phytotoxic activity of **1** against weed species has been recognized [39], and **1** has also been reported to show antifungal activity against *Trichophyton rubrum* and *Aspergillus fumigatus* [40].

As a result of the isolation of two metabolites, particularly *p*-hydroxybenzoic acid (**2**), it was decided to investigate the production of the two metabolites during the culturing of the fungus. The HPLC chromatographic profile of the medium polarity fraction of the extract of *A. dauci* showed the peaks corresponding to *α*-acetylorcinol (**1**) and *p*-hydroxybenzoic acid (**2**) at *t*_R_ 6.5 and 7.4 min, respectively (Figure 4). Additionally, the HPLC chromatographic profile of the *A. dauci* crude extract showed the presence of several minor components identified as the diketopiperazines cyclo-(pro-val) (**3**), cyclo-(leu-tyr) (**4**), cyclo-(pro-leu) (**5**), cyclo-(pro-phe) (**6**), cyclo-(val-leu) (**7**), cyclo-(val-phe) (**8**), and cyclo-(leu-phe) (**9**) (Figure 5), previously reported from a number of microorganisms and recognized as an important group of bioactive secondary metabolites [42].

The HPLC analyses of the medium polarity fractions obtained from extracts of cultures of *A. dauci* kept at different times showed the presence of **2** within 6 h of initiating the culture of the pathogen (Table 1; Appendix A), its production reached a maximum (42.9 ± 1.0 µg/mg) after 72 h. Alternatively, the production of *α*-acetylorcinol (**1**) in cultures of *A. dauci* was first detected after 48 h, with its maximum production (160.2 ± 3.8 µg/mg) occurring after 96 h of culturing the fungus (Table 1; Appendix A). These results indicate that *p*-hydroxybenzoic acid (**2**) is one of the first metabolites produced by *A. dauci* in the liquid culture conditions used in this investigation. These findings are of particular interest as it has been reported that fungal metabolites produced during spore germination or in the early stages of fungal growth play a significant role in plant–pathogen interaction [43]. Furthermore, the phytotoxic effect of *p*-hydroxybenzoic acid (**2**) against different plant species, in different assay models, supports it being involved in fungal pathogenicity.

## 3. Materials and Methods

### 3.1. General Experimental Procedures

The IR spectra were obtained on a Thermo Scientific Nicolet 6700 spectrometer (Waltham, MA, USA), dissolving the samples in CHCl_3_. Nuclear Magnetic Resonance (NMR) spectra were recorded on a Varian/Agilent spectrometer (AR Premium COMPACT; Santa Clara, CA, USA) at 600 (^1^H) and 150 MHz (^13^C); chemical shifts are listed as *δ* values. Gas chromatography–mass spectrometry analyses were carried out using an Agilent (Santa Clara, CA, USA) 6890N gas chromatograph coupled to an Agilent 5975C INERT mass spectrometer detector, using an electron impact (70 eV) ionization source. HPLC analyses were carried out using a Waters Alliance^®®^ (Milford, MA, USA) HPLC system equipped with a 2998-photodiode array detector (PDA). Control of the equipment, data acquisition, processing, and management of chromatography was performed by the Empower 2 software program (Waters) (version 3). Analytical HPLC was performed using a Phenomenex–Luna (Torrance, CA, USA) RP–C18 column (150 × 4.6 mm, 5 µm particle size), with a flow rate of 1.0 mL/min. Column chromatography (CC) was carried out on Silica gel 60 Å (70-230 mesh ASTM) (E.M. Merck. Darmstadt, Germany). Thin-layer chromatography (TLC) analyses were carried out on silica gel 60 F254 plates (E.M. Merck. Darmstadt, Germany) using phosphomolybdic acid as a visualizing reagent. Deionized water for HPLC was obtained using a Simplicity^®®^ Water Purification System (Millipore Sigma). All reagents and solvents were obtained from Sigma-Aldrich-Fluka Chemicals (St Louis, MO, USA).

### 3.2. Fungal Material

The strain of *A. dauci* (Kiihn) Groves and Skolko (FRA017; isolated at IRHS and deposited in COMIC (collection number COMIC C0003), France) was maintained in water/glycerol (20:80) at room temperature. A portion of the mycelium was used to inoculate V8 agar-containing Petri dishes, where the fungus was allowed to grow for 15 days at 25 °C under natural light/darkness conditions. The plates were then flooded with sterile distilled water and gently rubbed with a sterile bent plastic rod to release spores. Fernbach flasks containing 1 L of Czapek-Dox liquid medium were inoculated with a conidial suspension adjusted to 1 × 10^4^ conidia/mL of *A. dauci*; cultures were kept under shake (100 rpm) conditions, at 25 °C, for 96 h. Czapek-Dox liquid medium contained the following compounds: 1.5 g of potassium nitrate (Sigma-Aldrich^®®^), 0.5 g of potassium chloride (Fermont), 0.5 g of magnesium sulfate (Sigma-Aldrich^®®^), 0.01 g of ferric sulfate (Sigma-Aldrich^®®^), 1.0 g of monobasic potassium phosphate (J.T. Baker^®®^), 6 g of sucrose (commercial brand), and 2.0 g of casein (Sigma Aldrich^®®^) in 1 L of distilled water, with a final pH of 6.0 [10,44].

### 3.3. Extraction, Isolation, and Purification of Phytotoxic Metabolites

The mycelium was separated from the culture filtrate by filtration through cheesecloth. The aqueous filtrate was then extracted with ethyl acetate (3×, 1:1, *υ*/*υ*) and the organic layer was evaporated *in vacuo* to produce 432.5 mg of extract (ca. 45 mg/L). A suspension of the organic crude extract (420 mg) in 50 mL of a mixture of H_2_O/MeOH (95:5, *υ*/*υ*) was successively partitioned with hexane (3×, 1:1, *υ*/*υ*) and ethyl acetate (3×, 1:1, *υ*/*υ*), to produce the hexane (**A**; 43.8 mg), ethyl acetate (**B**; 202.7 mg), and aqueous (**C**; 132.6 mg) fractions. Fraction **B** was purified by column chromatography (30 × 2.5 cm) using a gradient elution with mixtures of hexane-dichloromethane-methanol to yield sixteen semi-purified fractions (**B1**–**B16**). Column chromatographic purification (30 × 1.0 cm) of phytotoxic fractions **B10** (14.2 mg) and **B11** (9.8 mg), using a gradient elution with mixtures of hexane and ethyl acetate, allowed the isolation of *α*-acetylorcinol (**1**, 10.2 mg) and *p*-hydroxybenzoic acid (**2**, 3.4 mg), respectively, in pure form.

*α-Acetylorcinol* (**1**): Amber oil; UV (MeOH) λ_max_ 208 and 279 nm; IR (CHCl_3_) ν_max_ 3377, 1697, 1600, and 1148 cm^−1^; ^1^H NMR (600 MHz, CD_3_OD) δ 6.16 (m, 3H), 3.54 (s, 2H), 2.12 (s, 3H). ^13^C NMR (150 MHz, CD_3_OD) δ 209.6, 159.8, 137.8, 108.9, 102.3, 51.7, 28.9 [13]; EIMS *m*/*z* 166 [M]^+^ (100), 123 (C_7_H_7_O_2_, 95).

*p-Hydroxybenzoic acid* (**2**): White powder; UV (MeOH) λ_max_ 207 and 253 nm; IR (CHCl_3_) *ν*_max_ 3465, 2922, 1673, 1593, 1251 cm^−1^; ^1^H NMR (600 MHz, CD_3_OD) δ 7.87 (dd, *J* = 8.4, 1.8 Hz, 2H), 6.81 (dd, *J* = 8.4, 1.8 Hz, 2H). ^13^C NMR (150 MHz, CD_3_OD) δ 170.6, 163.1, 132.9, 123.4, 115.9 [14,15]; EIMS *m*/*z* 138 [M]^+^ (58), 121 (C_7_H_5_O_2_, 100), 93 (C_6_H_5_O, 25).

### 3.4. Preparation of O,O-Dimethyl-α-acetylorcinol

A solution of *α*-acetylorcinol (**1**, 7.5 mg) and K_2_CO_3_ (100 mg) in dry acetone (1 mL) was treated with iodomethane (0.5 mL) and the reaction mixture was allowed to stir overnight at room temperature. The mixture was diluted with water and the resulting aqueous suspension was extracted with dichloromethane; the organic layer was dried over Na_2_SO_4_, filtered, and the solvent evaporated in vacuo to yield 8.6 mg (98%) of crude product identified as *O,O-*dimethyl-*α*-acetylorcinol: Amber oil; UV (MeOH) λ_max_ 210 and 275 nm; IR (CHCl_3_) ν_max_ 2938, 1712, 1596, 1463, 1200, and 1157 cm^−1^; ^1^H NMR (600 MHz, CDCl_3_) δ 6.37 (t, *J* = 2.4 Hz, 1H), 6.35 (d, *J* = 2.4 Hz, 2H), 3.78 (s, 6H), 3.61 (d, *J* = 0.6 Hz, 2H), 2.15 (d, *J* = 0.6 Hz, 3H); EIMS *m*/*z* 194 [M]^+^ (90), 151 (C_9_H_11_O_2_, 100).

### 3.5. Quantification of α-Acetylorcinol and p-Hydroxybenzoic Acid in Organic Fractions

#### 3.5.1. Culturing of a Dauci for Quantitative Analyses

A 1 mL spore suspension (adjusted to 1 × 10^4^ conidia/mL of *A. dauci*) was used to inoculate 100 mL of Czapek-Dox culture medium contained in an Erlenmeyer flask. Cultures were allowed to grow at 25 °C for different lengths of time (6, 12, 24, 48, 72, 96, and 168 h). The mycelium was separated from the liquid medium culture by filtration through cheesecloth, and the aqueous filtrate was extracted with ethyl acetate (3×, 1:1, *υ*/*υ*). The organic layer was dried over sodium sulfate, filtered, and the solvent evaporated in vacuo to produce the corresponding organic extract (ca. 2.0 mg/100 mL).

#### 3.5.2. Preparation of Standard Solutions for HPLC Analyses

Stock standard solutions of *α*-acetylorcinol (**1**) and *p*-hydroxybenzoic acid (**2**) were prepared at a concentration of 1 mg/mL in acetonitrile. Five standard working solutions (0.1 to 0.5 mg/mL) of each phytotoxin were prepared by serial dilution of the standard stock solutions with acetonitrile and used for the preparation of the corresponding calibration curves. All solutions were filtered through 0.45 mm membrane filters before injection. Under the conditions described, the retention times (*t*_R_) of *α*-acetylorcinol (**1**) and *p*-hydroxybenzoic acid (**2**) were found to be 6.5 and 7.4 min, respectively. The location of each phytotoxic metabolite in the chromatographic profiles was confirmed by the co-injection of extracts with the pure metabolites.

#### 3.5.3. Sample Preparation for HPLC Analyses

Samples of *A. dauci* organic extracts (ca. 2 mg) were suspended in 300 µL of a 9:1 mixture of water-methanol. The resulting suspension was extracted with ethyl acetate (3×, 1:1, *υ*/*υ*), and the organic layer was dried over sodium sulfate and filtered. The solvent evaporated to produce the corresponding ethyl acetate fractions, which were prepared at a concentration of 1 mg/mL in acetonitrile. All solutions were filtered through 0.45 mm membrane filters before injection.

#### 3.5.4. Instrumentation and Conditions for HPLC Quantification of Phytotoxic Metabolites

Quantitative analyses by HPLC were performed using a Waters Alliance^®®^ HPLC instrument equipped with a quaternary pump, degasser, autosampler, and 2998-photodiode array detector (PDA). Control of the equipment, data acquisition, processing, and management of the chromatograph was carried out using the Empower 2 software program (Waters) (version 3). All analyses were carried out on a reverse-phase Luna RP–C18 column (150 × 4.6 mm, 5 µm particle size). Chromatographic conditions included a mobile phase of A: Acetonitrile and B: 0.1% (*v*/*v*) formic acid in water, following a gradient elution program of 10−50% A and 90−50% for 0−40 min; a flow rate of 1.0 mL/min, room temperature, and an injection volume of 10 µL. The UV detector was set to wavelengths of 210 and 257 nm.

#### 3.5.5. Method Validation

The HPLC-PDA method was validated according to the International Conference on Harmonization Guidelines (ICH: Validation of Analytical Procedures) [45]. Standard calibration curves for quantifying the two phytotoxins were obtained by plotting concentration (mg/mL) against response. Five and six different concentrations for each phytotoxin, in the range of 0.1–0.5 mg/mL for *α*-acetylorcinol (**1**) and 0.05–0.5 mg/mL for *p*-hydroxybenzoic acid (**2**), were prepared in triplicate. The linearity of these curves was evaluated by the least-squares fit and the correlation coefficient. The detection wavelengths were 210 nm for *α*-acetylorcinol (**1**) and 257 nm for *p*-hydroxybenzoic acid (**2**). The limit of detection (LOD) and limit of quantification (LOQ) were determined based on the standard deviation (σ) of the response and the slope (*S*) (Appendix A):LOD = 3.3 × σ/*S*
LQD = 10 × σ/*S*

The precision was evaluated using repeatability (intraday) and reproducibility (interday). Intraday and interday were evaluated by analyzing six replicate injections of the two phytotoxins on one and two different days. The relative standard deviation (RSD) was calculated for each determination and taken as a measure of precision.

Percent recovery was calculated in order to evaluate the accuracy of the method. The sample matrix (*A. dauci* organic fraction) was spiked with known amounts of phytotoxins at different concentrations levels: *α*-Acetylorcinol (**1**) (0.2, 0.3, and 0.4 mg/mL) and *p*-hydroxybenzoic acid (**2**) (0.05, 0.1, and 0.2 mg/mL). Each sample was analyzed in sextuplicate (Appendix A).

Data are expressed as the means ± SD for the number of samples at each time (*n* = 3). Analysis of variance (ANOVA) was used to analyze the content of phytotoxins in *A. dauci* organic extract cultured at different times. Tukey’s post-hoc test was used to determine the significant differences; *p* < 0.05 was considered statistically significant. Prisma Graph-Pad software (version 4.0) was used for statistical analysis.

### 3.6. Phytotoxic Assays

#### 3.6.1. Leaf-Spot Assay

The assay was carried out as previously described [46]. Parsley (*Petroselium crispum*) leaves were excised and washed with sterilized water. A 20 µL drop of the sample (5.0 mg/mL for extracts and fractions and 7.25 mM for purified phytotoxins), non-inoculated medium (negative control), or sterile distilled water (blank), was placed over a lightly scratched area made with a sterile, trimmed paintbrush on the adaxial face of the leaf. The leaves were kept in a high-humidity chamber under natural light at 25 °C for 72 h. Three leaves, with two applications each, were used for each treatment.

#### 3.6.2. Leaf Infiltration Assay

The assay was carried out as previously described [47,48]. Ten microliters of extract, fractions (5.0, 7.5, and 10 mg/mL), or purified phytotoxins (7.25, 14.5, 22.0 mM) was infiltrated abaxially into leaves of tobacco (*Nicotiana alata*) and marigold (*Tagetes erecta*) (both 4–6 weeks old) plants. Distilled water and non-inoculated medium were used as controls. All evaluations were made in triplicate. Lesion formation was documented after 24 h.

#### 3.6.3. Leaf Immersion Assay

The assay was carried out as previously described [49]. The basal part of the stem, of parsnip (*Pastinaca sativa*) and parsley (*P. crispum*) leaves, was placed in a glass tube containing 1 mL of a solution prepared by combining 100 µL of the test solution (5.0, 7.5, and 10 mg/mL for extracts and fractions, and 7.25, 14.5, and 22.0 mM for purified phytotoxins) and 900 µL of water. The leaf was kept in a high-humidity chamber under natural light conditions at 25 °C. Distilled water and non-inoculated medium were used as controls. All evaluations were made in triplicate. The phytotoxic effect was recorded after 48 h.

## 4. Conclusions

Two metabolites, *α*–acetylorcinol (**1**) and *p*-hydroxybenzoic acid (**2**), have been identified in the cultures of *A. dauci* grown in Czapek-Dox medium. The results of the evaluation of the phytotoxic activity of the two metabolites against different plant species, in different assay models, showed that while *α*–acetylorcinol (**1**) was weakly phytotoxic, *p*-hydroxybenzoic acid (**2**) caused a phytotoxic effect in different plant species. Furthermore, the production of **2** during the early stages of fungal growth suggests that this metabolite might play an important role in the plant–pathogen interaction between *A. dauci* and *D. carota*.

## Figures and Tables

**Figure 1 molecules-25-04003-f001:**
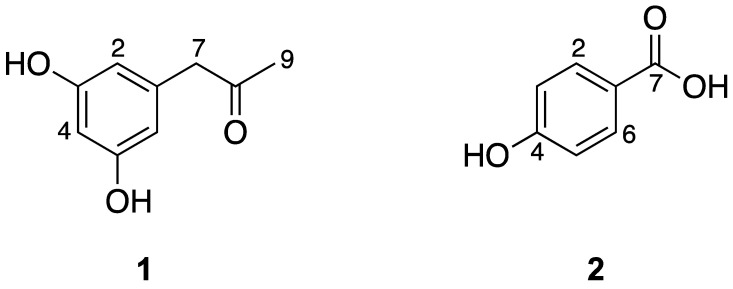
Phytotoxic secondary metabolites isolated from the medium polarity fraction of *A. dauci*; *α*-acetylorcinol (**1**) and *p*-hydroxybenzoic acid (**2**).

**Figure 2 molecules-25-04003-f002:**
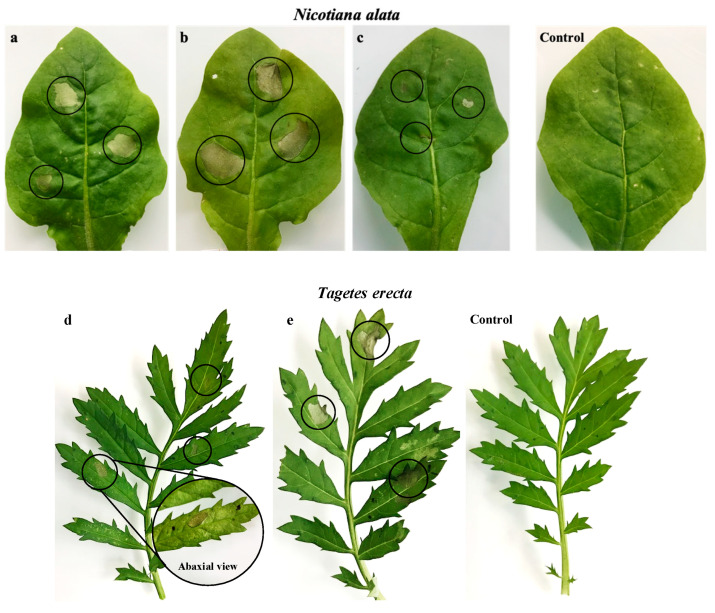
Leaf infiltration assay. Phytotoxic damage of *α*-acetylorcinol (**1**) and *p*-hydroxybenzoic acid (**2**) on tobacco (*N. alata*) and marigold (*T. erecta*) leaves: (**a**) *p*-hydroxybenzoic acid (14.5 mM), (**b**) *p*-hydroxybenzoic acid (22.0 mM), (**c**) *α*-acetylorcinol (22.0 mM), (**d**) *p*-hydroxybenzoic acid (14.5 mM), and (**e**) *p*-hydroxybenzoic acid (22.0 mM). The sterile uninoculated medium was used as a control. The circle indicates the phytotoxic effect.

**Figure 3 molecules-25-04003-f003:**
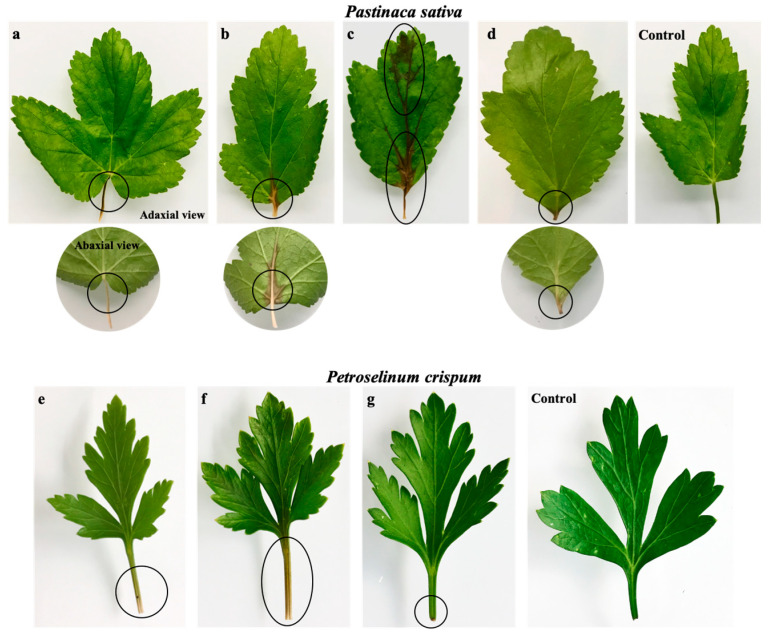
Leaf immersion assay. Phytotoxic damage of *α*-acetylorcinol (**1**) and *p*-hydroxybenzoic acid (**2**) on leaves excised from parsnip (*P. sativa*) and parsley (*P. crispum*) plants: (**a**) *p*-hydroxybenzoic acid (7.25 mM), (**b**) *p*-hydroxybenzoic acid (14.5 mM), (**c**) *p*-hydroxybenzoic acid (22.0 mM), (**d**) *α*-acetylorcinol (22.0 mM), (**e**) *p*-hydroxybenzoic acid (14.5 mM), (**f**) *p*-hydroxybenzoic acid (22.0 mM), and (**g**) *α*-acetylorcinol (22.0 mM). The sterile uninoculated medium was used as a control. The circle indicates the phytotoxic effect.

**Figure 4 molecules-25-04003-f004:**
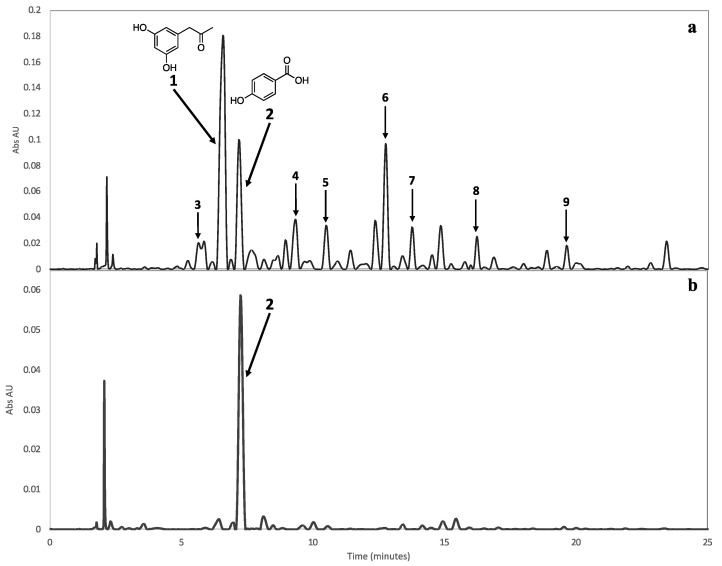
HPLC chromatographic profile of the medium polarity fraction from the extract of *A. dauci* at 96 h of culture. The chromatogram shows the peaks corresponding to *α*-acetylorcinol (**1**), *p*-hydroxybenzoic acid (**2**), and the diketopiperazines (**3**–**9**) at (**a**) 210 and (**b**) 257 nm.

**Figure 5 molecules-25-04003-f005:**
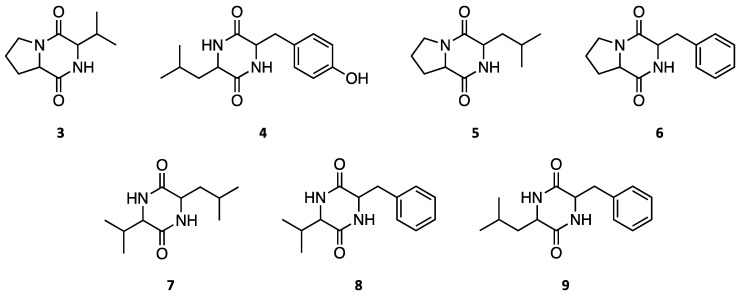
Minor diketopiperazines identified in the crude extract of *A. dauci*; cyclo-(pro-val) (**3**), cyclo-(leu-tyr) (**4**), cyclo-(pro-leu) (**5**), cyclo-(pro-phe) (**6**), cyclo-(val-leu) (**7**), cyclo-(val-phe) (**8**), and cyclo-(leu-phe) (**9**).

**Table 1 molecules-25-04003-t001:** Quantification of phytotoxins **1** and **2** in organic extracts from cultures of *A. dauci* kept at different times of culture.

Time of Culture (h)	Content (µg/mg Extract) (Mean ± SD)	
	*α*-Acetylorcinol (1)	*p*-Hydroxybenzoic Acid (2)
6	-	31.9 ± 0.2
12	-	32.0 ± 0.5
24	-	33.6 ± 1.0
48	90.8 ± 3.4	41.7 ± 1.3
72	124.3 ±1.7	42.9 ± 1.0
96	160.2 ± 3.8	41.0 ± 0.2
168	154.0 ± 4.9	30.5 ± 0.3

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
