# Peer review of "Identification and Quantification of a Phytotoxic Metabolite from Alternaria dauci"

_molecules, 2020, doi:10.3390/molecules25174003_

Round 1
Reviewer 1 Report
The manuscript needs corrections:
Lines 264, 271, 277: “nM” should be corrected to “mM”.
Author Response
The term “mM” has been corrected in Lines 281, 288 and 294
Reviewer 2 Report
The authors should cite paper from Journal of the Mexican Chemical Society on diketopiperazienes. These compounds should also be marked in chromatograms recorded by to authors. Please provide more data on chemical composition and discuss them.
Author Response
We have included the citation of the manuscript recently accepted for publication in the Journal of the Mexican Chemical Society Paper (Lines 137 and 437). We have also indicated the presence of the different diketopiperazines in the HPLC chromatographic profile of the crude extract of A. dauci (Figure 4a) and have included a new figure with the structures for the seven minor metabolites (Figure 5). Finally, we added a brief comment about the importance of diketopiperazines (lines 133-137).
Round 2
Reviewer 2 Report
No further comments.
This manuscript is a resubmission of an earlier submission. The following is a list of the peer review reports and author responses from that submission.
Round 1
Reviewer 1 Report
The authors report on the isolation of two trivial metabolites from Alternaria dauci with known phytotoxic effects. Novel results are some data on toxic effects on several plants, and the monitoring of occurrance of these metabolites over time using a validated HPLC method. So this manuscript is of limited novelty, and suitable for publication in Molecules only after major revision, especially significant improvement of the part on quantification of the metabolites.
Concerns:
Line 99: Please discuss why p-hydroxybenzoic acid, a compound previously isolated from carrots, is at the same time a phytotoxin for carrots (see line 13).
Line 126, Table 1, and others: numerous values are presented with undue precision (e.g., 42.87 ± 0.95 μg/mg). Please round the numbers in order to present believable values.
Chapter 3.3. Extraction: a yield of 1.2 mg of acetylresorcinol is described here, but in 3.4. Preparation of O,O-dimethyl-α-acetylorcinol a much larger amount of 7.5 mg is used for derivatization. This is inconsistent! Or did the authors use synthetic or commercial product for the biological investigations?
Line 183, NMR data: 6.5 ppm, “singlet” for 3 aromatic H is not correct, since the NMR in the Sup. Information clearly shows that this is not a singlet. 13C NMR for this compound is missing. Reference to published NMR data of this compound is missing.
Concerning validation of the analytical method (Table S1): it is simply impossible that the value for LOD is higher that the value for LOQ! Values for linearity and LOD/LOQ should be presented with the same units (mg per mL versus micrograms per mL).
Following Table S1 LOD/LOQ were determined via singnal-to-noise relationship, but in the core manuscript another method is mentioned (“The limit of detection (LOD) and limit of quantification (LOQ) were determined based on the 239 standard deviation (σ) of the response and the slope (S)”). This is not consistent. Further, the linear ranges for both analytes are really small! Please comment on that.
Others:
p-hydroxybenzoic acid is not a salicyclic acid derivative;
control usage of capital letters in headings of chapters;
line 183: 6.15 (s, 3H9) ??;
line 214: source of “authentic samples”?
Reviewer 2 Report
This manuscript presents the identification and quantification of two phytotoxic metabolites from Alternaria dauci. The work seems carefully done and the introduction, experiment design, analyses, results, and conclusions are clearly presented.
However, the concentrations of samples used to phytotoxic assay are too high (14.5 and 22.0 mM). In addition, all the metabolites are known compounds and the phytotoxicity of p-hydroxybenzoic acid is also known. From these reasons, this reviewer feels that the manuscript is marginal to be published in this journal.
Reviewer 3 Report
The paper submitted by Leyte-Lugo et al. is focused on the identification of phototoxic compounds from Alternaria species in plants. In general, the paper is very interesting and all experiments are correctly performed. in my opinion, the manuscript should be published in Molecules after addressing one comment given below.
The authors identified and tested only two metabolites among others seen on HPLC chromatogram. Do the authors have an idea what other compounds may be produced by Alternaria and whether they also can affect plants as two identified compounds? Is it possible to characterize other compounds seen in Fig 4 by providing retention time and UV-Vis maxima as supplementary material? By the way, it would be more clear for the reader to present Fig. 4 in a different manner so both chromatograms are clearly seen (please prepare one figure showing both wavelengths in separate rows one below the other).